Genome-wide identification and expression analyses of SWEET gene family reveal potential roles in plant development, fruit ripening and abiotic stress responses in cranberry (Vaccinium macrocarpon Ait)

Chen Li 1
Cai Mingyu 1
Liu Jiaxin 1
Jiang Xuxin 1
Liu Jiayi 1
Zhenxing Wang zhenxinghd@aliyun.com 1
Wang Yunpeng wangyunpengcas@163.com 2
Li Yadong 1
1 Jilin Agricultural University, College of Horticulture , Changchun , China
2 Institute of Agricultural Biotechnology, Jilin Academy of Agricultural Sciences , Changchun , China
Eamens Andrew
Electronic publication date: 2024 Sep 19
Publication date: 2024
Volume: 12
Electronic Location ID: e17974
Received 2023 Dec 7; Accepted 2024 Aug 5
Copyright: ©2024 Chen et al.
Copyright year: 2024
Copyright holder: Chen et al.
License: This is an open access article distributed under the terms of the Creative Commons Attribution License, which permits unrestricted use, distribution, reproduction and adaptation in any medium and for any purpose provided that it is properly attributed. For attribution, the original author(s), title, publication source (PeerJ) and either DOI or URL of the article must be cited.
License URL: https://creativecommons.org/licenses/by/4.0/

Keywords: Cranberry, SWEET, Bioinformatics analysis, Expression analysis, Growth and development, Abiotic stress

Funding: JiLin Provincial Natural Science Foundation of China 202101013697JC National college student innovation training program 202310193041 JiLin Provincial Development and Reform Commission Project 2023C0354-4 JiLin Province Science and Technology Development Plan Project 20220208099RC This work was supported by grants from JiLin Provincial Natural Science Foundation of China (202101013697JC); National college student innovation training program (202310193041); JiLin Provincial Development and Reform Commission Project (2023C0354-4); JiLin Province Science and Technology Development Plan Project (20220208099RC). The funders had no role in study design, data collection and analysis, decision to publish, or preparation of the manuscript.

==============================
The sugars will eventually be exported transporter (SWEET) family is a novel class of sugar transporters that play a crucial role in plant growth, development, and responses to stress. Cranberry (Vaccinium macrocarpon Ait.) is a nutritious berry with economic importance, but little is known about SWEET gene family functions in this small fruit. In this research, 13 VmSWEET genes belonging to four clades were identified in the cranberry genome for the first time. In the conserved domains, we observed seven phosphorylation sites and four amino acid residues that might be crucial for the binding function. The majority of VmSWEET genes in each clade shared similar gene structures and conserved motifs, showing that the VmSWEET genes were highly conserved during evolution. Chromosomal localization and duplication analyses showed that VmSWEET genes were unevenly distributed in eight chromosomes and two pairs of them displayed synteny. A total of 79 cis-acting elements were predicted in the promoter regions of VmSWEETs including elements responsive to plant hormones, light, growth and development and stress responses. qRT-PCR analysis showed that VmSWEET10.1 was highly expressed in flowers, VmSWEET16 was highly expressed in upright and runner stems, and VmSWEET3 was highly expressed in the leaves of both types of stems. In fruit, the expression of VmSWEET14 and VmSWEET16 was highest of all members during the young fruit stage and were downregulated as fruit matured. The expression of VmSWEET4 was higher during later developmental stages than earlier developmental stages. Furthermore, qRT-PCR results revealed a significant up-regulation of VmSWEET10.2, under osmotic, saline, salt-alkali, and aluminum stress conditions, suggesting it has a crucial role in mediating plant responses to various environmental stresses. Overall, these results provide new insights into the characteristics and evolution of VmSWEET genes. Moreover, the candidate VmSWEET genes involved in the growth, development and abiotic stress responses can be used for molecular breeding to improve cranberry fruit quality and abiotic stress resistance.

Introduction

Sugars are important molecules that regulate a wide range of morphological and physiological processes in plants. Apart from their functions as energy sources, osmoregulators, storage molecules, and structural components, sugars also act as signaling molecules that interact with diverse plant signaling pathways including hormones, stress responses, and light perception mechanisms. Consequently, sugars modulate growth and development in response to dynamic environmental conditions (Mishra, Sharma & Laxmi, 2022). As the primary photoassimilate, sugars are synthesized in leaves before being transported via the phloem to sink tissues, such as flowers, stems, tubers, swollen tap roots, fruits, and seeds (Sonnewald & Fernie, 2018). Phloem loading in source leaves and unloading in sink tissues involves a combination of the symplastic, apoplastic, and/or polymer trapping pathways. The symplastic and polymer trapping pathways are passive processes that are correlated with source activity and sugar gradients. In contrast, the apoplastic pathway is characterized by active energy consumption, which necessitates the involvement of sugar transporters for efficient translocation of sugars (De Schepper et al., 2013). In higher plants, three sugar transporter families play a crucial role in phloem loading and unloading: the monosaccharide transporter-like (MST) gene family, the sucrose transporters (SUT/SUC), and sugars will eventually be exported transporters (SWEET) (Doidy et al., 2012). The MSTs and SUTs contain 12 transmembrane domains (TMDs) and require energy to complete the transmembrane transport of sugars. However, SWEETs have seven TMDs and act as uniporters that facilitate sugar translocation along a concentration gradient independently of the proton gradient and pH (Chen et al., 2010; Chen, 2014; Yuan & Wang, 2013; Julius et al., 2017). To date, SWEET genes have been identified in grain, horticultural, legume, oil and fiber crops and other plant species, such as wheat (Gao et al., 2018), soybean (Patil et al., 2015); oilseed rape (Jian et al., 2016), cotton (Li et al., 2018), apple (Wei et al., 2014), jujube (Yang et al., 2023), tomato (Feng et al., 2015), cabbage (Zhang et al., 2019a), daylily (Huang et al., 2022), and saccharum (Hu et al., 2018). Phylogenetically, plant SWEETs are divided into four clades (Clades I, II, III and IV) based on the functional characterization of SWEET genes in Arabidopsis thaliana (Chen et al., 2010). Clades I, II, and IV tend to transport monosaccharides, and Clade III predominantly transports sucrose, a disaccharide (Le Hir et al., 2015). Additionally, the Clade IV members are typically localized to the tonoplast (Chardon et al., 2013; Klemens et al., 2013), while members of other clades are situated primarily on the plasma membrane and sometimes on the Golgi membrane and chloroplast (Breia et al., 2021).

Passive unloading of sucrose from the mesophyll into the apoplast and its subsequent active loading into the phloem have been described by Giaquinta in the late 1970s (Giaquinta, 1977), but the mechanism underlying this unloading process remained elusive until the discovery of SWEET genes. In Arabidopsis, AtSWEET11 and AtSWEET12 were found to encode proteins that facilitate the release of sucrose from parenchyma cells to the apoplast, and the atsweet11;12 double mutation suppressed phloem transport, which led to the accumulation of starch in the leaves (Chen et al., 2012). Recent studies have revealed that SWEET-mediated phloem loading in leaves is regulated by sugar signals. In Chinese jujube, the transcription of ZjSWEET2.2 was activated by a low sugar signal, while expression decreased and the photosynthetic rate was reduced by a high sugar signal (Geng, Wu & Zhang, 2020). In addition, multiple physiological functions of SWEET transporters including nectar secretion, pollen nutrition, grain filling, fruit ripening, shoot branching and bud outgrowth have been reported (Eom, Chen & Sosso, 2015; Wen et al., 2022; Gautam et al., 2019). In Arabidopsis, Brassica rapa and tobacco, the SWEET9 gene was identified in the transport of sucrose from nectary parenchyma to the extracellular space for rewarding pollinators, and atsweet9 mutant lines failed to nectar secretion (Lin et al., 2014). In maize and rice, SWEET genes play a role in the transfer of sugars imported from the maternal phloem. Notably, mutants zmsweet4c, ossweet4, and ossweet11, as well as the ossweet11;15 double mutants, exhibited significantly decreased sucrose concentration in the embryo, accumulated starch in the pericarp, and were deficient in seed filling (Sosso et al., 2015; Ma et al., 2017; Yang et al., 2018). In pineapple, AcSWEET11 was strongly expressed in ripening fruit, and the overexpression of AcSWEET11 in the pineapple callus and in tomato enhanced sugar content (Lin et al., 2022). In tomato, mutation of the SlSWEET15 gene resulted in a significant reduction in the average size and weight of fruits and was accompanied by severe impairments in seed filling and embryo development (Ko et al., 2021). The above results indicate that SWEET proteins mediate the unloading of sucrose in sink organs that affect the yield and quality of important economic crops.

Sugar transport and partitioning not only affect plant growth and development, but also respond to abiotic and biotic stress. As SWEETs facilitate the efflux of sugars, they are highly susceptible to being hijacked by pathogens, making them central players in plant-pathogen interactions. In Arabidopsis, the root tonoplast-localized AtSWEET2 was induced during Pythium irregulare infection, which led to enhanced cytosolic sugar accumulation in the vacuole. Overexpression of AtSWEET2 enhanced plant resistance to P. irregulare by limiting sugar availability to the pathogen (Breia et al., 2021). However, the opposite behavior was observed in grapes. Overexpression of VvSWEET4 improved resistance to P. irregulare, while high sugar accumulation in hairy roots provided better support for increased energy demand during pathogen infection (Meteier et al., 2019). Thus, it is difficult to define roles for SWEET transporters in plant-pathogen interactions, because the metabolic signatures and regulatory nodes that determine susceptibility or resistance responses remain poorly understood. Previous studies on the response of SWEET transporters to abiotic stress focused on drought, cold, and salinity. In Fen Jiao M. acuminata AAB group which exhibits high tolerance to abiotic stresses resistant, some MaSWEETs exhibited increased expression in response to cold, drought, salinity, and fungal disease (Miao et al., 2017). In tea (Camellia sinensis), the tonoplast sugar transporter gene CsSWEET16 was downregulated under cold stress. Overexpression of CsSWEET16 in Arabidopsis resulted in enhanced cold tolerance, which was accompanied by glucose accumulation in the vacuole and reduced levels of fructose (Wang et al., 2018). Although our understanding of SWEET functions is increasing, their roles in sugar transport, distribution, metabolism, and signaling require further study.

Cranberry (Vaccinium macrocarpon Ait.), a diploid plant (2n = 2x = 24), is a woody perennial belonging to the Ericaceae family w’ithin the Vaccinium genus (Kron et al., 2002). It is endemic to North America and can also be found in the Changbai Mountains of northeastern China. Like other members of its botanical family, such as blueberry, bilberry, and lingonberry, cranberry is uniquely adapted to life in cool and moist peat bogs; it thrives in acidic, nutrient poor soils (Fajardo et al., 2012). This small but economically important berry fruit offers immense potential for global development due to its versatility. The growing importance of cranberries has created a demand for enhanced productivity and superior quality. However, during commercial cultivation, cranberries frequently encounter abiotic stress including extreme temperatures (i.e., frost damage and heat stress) and water availability (both drought and flooding) due to disparities between cultivation the environment and their native habitat (Neyhart et al., 2022). Plant SWEET transporters have been demonstrated to play important roles in growth, development, and plant-environment interactions in many species, but systematic studies on SWEET genes in cranberry have not been reported. In this study, we conducted a genome-wide analysis of SWEET genes in cranberry and analyzed their phylogenetic relationship, gene structure, motif distribution, chromosomal localization, and cis-regulatory elements. In addition, spatiotemporal expression, and abiotic stress responses were carried out using qRT-PCR. This study provides valuable insights into the roles of VmSWEET genes in cranberry growth, development, and stress responses.

Materials and Methods

Plant materials

The cultivar Bain 11, in the small berry germplasm resource garden of Jilin Agricultural University (43°48′05″N, 125°24′15″E), was used to detect the expression of VmSWEET genes in cranberry tissues and fruit at different stages of development (Fig. 1). The average annual precipitation in this area is 600–700 mm, with an average temperature of 4.6 °C and an annual frost-free period lasting 140–150 d. In order to improve the garden soil for optimal cranberry growth, it was mixed with sand, peatmoss, perlite, and sulfur powder. The pH of the improved soil was 5.0, which is conducive to the successful cultivation of cranberries. Roots, upright stems, leaves of upright stems, runner stems, leaves of runner stems, and flowers were collected during the flowering period. Fruits at young stage (S1), expansion stage (S2), color turning stage (S3), and maturity stage (S4) were collected 10, 30, 60, and 80 d after full bloom, respectively. Tissue and fruit samples were randomly obtained from three plants and replicated three times.

Figure 1 Different cranberry tissues and fruit at different stages of development.

Plantlets of Bain11 were used to detect expression patterns under abiotic stress. An osmotic treatment (20% PEG 8000), saline treatment (200 mM NaCl), saline-alkaline treatment (30 mM Na2CO3 and 30 mM NaHCO3), and aluminum treatment (5mM AlCl3) were administered by immersing the roots of plantlets in containers with different solutions. Tender stem apexes were collected 0, 3, 6, 9, 12, and 24 h during the different stress treatments. Samples were collected from three containers every time and replicated thrice. All samples were immediately frozen in liquid nitrogen and stored at −80 °C.

Identification SWEET gene family in cranberry

Members of the cranberry SWEET gene family were identified by protein Blast of the 17 Arabidopsis SWEET proteins against the V. macrocarpon genome database (https://www.ncbi.nlm.nih.gov/genome/?term=cranberry). The coding domain sequences (CDS) of VmSWEET genes are shown in File S1. The NCBI CDD (https://www.ncbi.nlm.nih.gov/Structure/cdd/wrpsb.cgi) and PFAM (http://pfam.xfam.org/) websites were used to search for the conserved domains of the candidate members.

Protein domain, conserved motifs, gene structure and promoter cis-regulatory elements analysis

The number of amino acids, molecular weights (MWs), and theoretical isoelectric points (PIs) were analyzed using the ExPASy website (https://web.expasy.org/protparam/). Subcellular localization of VmSWEETs was predicted using WoLFPSORT (https://www.genscript.com/wolf-psort.html). A more recent and better transmembrane predictor TMHMM 2.0 (https://services.healthtech.dtu.dk/services/TMHMM-2.0/) was used for prediction of TMDs. The conserved motifs of SWEETs were predicted using online MEME tools (https://meme-suite.org/meme/tools/meme). The MEME parameter settings were as follows: the number of motifs was 10 and the range of motifs varied from 5–50. The exon/intron structures were analyzed using TBtools software (South China Agricultural University, Guangzhou, China, https://github.com/CJ-Chen/TBtools/releases) (Chen et al., 2020). Promoter cis-acting regulatory elements were predicted by submitting 2 kb upstream sequences of the translation start site of VmSWEET genes to the PlantCARE web site (http://bioinformatics.psb.ugent.be/webtools/plantcare/html). The promoter sequences and cis-acting elements of VmSWEET genes are shown in File S2.

Phylogenetic analyses and multiple sequence alignment

The amino acid sequences of 17 Arabidopsis thaliana SWEET genes, 21 Oryza sativa SWEET genes, 14 Vitis vinifera SWEET genes and 13 Vaccinium macrocarpon SWEET genes were used to construct an unrooted phylogenetic tree using MEGA 7.0 sofware (Mega Limited, Auckland, New Zealand, http://www.megasoftware.net) (Kumar et al., 2016) by the maximum likelihood method with bootstrap values for 1,000 replicates. Then the phylogenetic tree was illustrated by ITOL v6 (https://itol.embl.de/). The amino acid sequences of SWEET proteins from Arabidopsis, rice and grape are shown in File S3. Alignment of SWEET protein sequences was performed using the ClustalX software (Trinity College Dublin, Ireland, UK) and phosphorylation sites were predicted by NetPhos 3.1 (https://services.healthtech.dtu.dk/services/NetPhos-3.1/). The GENEDOC 3.20 software (http://nrbsc.org/gfx/genedoc) was used to highlight conserved or similar amino acid sequences.

Chromosomal distribution and gene synteny analysis

MapChart (http://www.joinmap.nl) (Voorrips, 2002) was used to construct the chromosomal distribution map of VmSWEET genes, and MCScanX (https://links.jianshu.com/go?to=https://github.com/wyp1125/MCScanX) was used to analyze gene synteny. First, potential homologous gene pairs in cranberry were identified by local all-vs-all BLASTp algorithm-based searches (E < 1e−10). Then, syntenic chains were identified by MCScanX using homologous pairs as input (Tang et al., 2008). Duplications in the SWEET gene family were identified by downstream analysis tools in the MCScanX package. Finally, synteny analysis was illustrated by CIRCOS.

Quantitative RT-PCR (qRT-PCR) for SWEET genes

Total RNA was isolated by a modified CTAB method (Li et al., 2019). The integrity and concentration of RNA were assessed using electrophoresis on 1.2% agarose gels and an IMPLEN P330 spectrophotometer (IMPLEN, Munich, Germany), respectively. A 1.0 µg sample of the extracted RNA was reverse transcribed into cDNA using a cDNA Synthesis SuperMix (TransGen Biotechnology, Beijing, China). qRT-PCR was performed with an ABI StepOne Plus Real-Time Quantitative PCR System (Applied Biosystems, Foster City, CA, USA) following the http://miqe.gene-quantification.info/ (the Minimum Information for Publication of Quantitative Real-Time PCR Experiments). SYBR Green I fluorescent dye as the detection signal was used for detection of VmSWEETs. The reaction procedure was as follows: denaturation at 94 °C for 30 s; denaturation at 94 °C for 5 s; annealing at 60 °C for 30 s; and then 94 °C for 10 s, 60 °C for 60 s, and 94 °C for 15 s to generate the melting curve. All experiments were run in three biological replicates. Primers were designed with the Primer-BLAST tool (http://blast.ncbi.nlm.nih.gov/). The VmSAND gene has been validated the optimal internal reference gene for analyzing various cranberry tissues and abiotic stress treatments (Li et al., 2019). It was utilized as a control to standardize the expression of VmSWEETs. The designed qRT-PCR primers are shown in File S4. The raw data for Ct values are shown in File S5. Relative quantitative analysis of 13 target genes in different cranberry tissues and fruit development stages was performed using the 2−ΔCt method, and column charts were obtained using SigmaPlot 10.0 (Systat Software, Inc., Melbourne). Expression profiles of VmSWEETs under abiotic stress were calculated using the 2−ΔΔCt method, and the expression levels were log2 transformed and normalized to obtain a heatmap by TBtools (South China Agricultural University, Guangzhou, China, https://github.com/CJ-Chen/TBtools/releases) (Chen et al., 2020). The MIQE checklist is shown in File S6. Statistical analysis was carried out by one-way ANOVA and LSD-test using the SPSS software (IBM Corporation, Armonk, NY, USA). A p value of less than 0.05 (p < 0.05) was considered statistically significant.

Results

Genome-wide identification and analysis of VmSWEET genes

Through homologous alignment and conservative domain verification, a total of 13 genes encoding SWEET proteins were identified. VmSWEETs were named based on their phylogenetic grouping into the four SWEET clades (Fig. 2), according to Doidy’s taxonomic framework (Doidy, Vidal & Lemoine, 2019). The physical and chemical details of VmSWEET genes are summarized in Table 1. The CDS lengths of VmSWEETs varied from 519–1065 bp, corresponding to amino acid numbers ranging from 196–354. The MWs of the 13 proteins ranged from 21.38–40.24 KD, and the PIs spanned from 6.24–9.61. The instability index ranged from 30.52–51.71, suggesting that 62% of VmSWEETs were hydrophobic. The aliphatic index of nearly all proteins exceeded 100, while the grand average of hydropathicity (GRAVY) values varied from 0.205–1.002 °C indicating their inherent hydrophobic properties. The transmembrane domains prediction using TMHMM suggested that VmSWEETs exhibit 5–7 TMDs. Subcellular localization prediction using WoLFPSORT revealed that VmSWEET1.1 and VmSWEET2.2 may be localized to the tonoplast membrane, VmSWEET1.2 was likely to be localized to the endoplasmic reticulum, and the other 10 VmSWEETs were primarily located in the plasma membrane of cell.

Figure 2 Phylogenetic analysis of the SWEET gene family in four species.

Different colors of the outer ring represent four different SWEET clades. Before the gene name, green triangles represent Arabidopsis thaliana, black boxes represent Oryza sativa, red dots represent Vitis vinifera, blue stars represent Vaccinium macrocarpon. The evolutionary history was inferred using the neighbor joining method with 1,000 replicates.

Table 1 Physical and chemical properties of SWEET genes in cranberry.

Gene name	Gene ID	CDS/bp	Protein length/aa	MWs/KD	PI	Instability index	Aliphatic index	GRAVY	TMDs	Predicted location(s)	
VmSWEET1.1	vmacro00890	741	246	26.91	9.2	30.53	106.95	0.628	7	TM	
VmSWEET1.2	vmacro05470	768	255	28.11	9.61	30.52	107.49	0.507	6	ER	
VmSWEET2.1	vmacro09417	702	233	26.25	8.87	47.52	120.86	0.861	7	PM	
VmSWEET2.2	vmacro03987	591	196	21.38	9.1	33.31	134.69	1.002	5	TM	
VmSWEET3	vmacro06571	1065	354	40.24	9.4	40.74	99.63	0.205	7	PM	
VmSWEET4	vmacro18238	774	257	28.72	8.83	37.28	115.64	0.597	7	PM	
VmSWEET5	vmacro19373	756	251	28.96	8.66	47.03	124.14	0.797	7	PM	
VmSWEET10.1	vmacro16733	879	292	32.48	8.56	38.24	128.53	0.861	7	PM	
VmSWEET10.2	vmacro19147	867	288	32.42	8.94	36.98	117.36	0.666	7	PM	
VmSWEET12	vmacro19148	807	268	30.41	9.44	39.65	111.31	0.512	7	PM	
VmSWEET13	vmacro16734	702	233	26.15	9.41	36.40	122.06	0.6	6	PM	
VmSWEET14	vmacro01036	948	315	35.09	6.24	51.71	109.84	0.369	5	PM	
VmSWEET16	vmacro08173	681	226	24.93	6.82	41.97	116.95	0.613	6	PM	
Notes.

CDS the length of coding domain sequences

MWs the molecular weight

PI theoretical isoelectric point

GRAVY grand average of hydropathicity

TMDs the number of transmembrane domains

PM plasma membrane

ER endoplasmic reticulum

TM tonoplast membrane

Phylogenetic analysis of putative VmSWEET proteins

To investigate the phylogenetic relationships among members of the SWEET gene family in cranberry and other plant species, a phylogenetic tree was constructed by aligning the predicted amino acids of 13 VmSWEET sequences, 17 AtSWEET sequences, 21 OsSWEET sequences, and 15 VvSWEET sequences. The 66 proteins were clustered into four different clades (Fig. 2). Clade I contained five VmSWEETs (VmSWEET1.1, 1.2, 2.1, 2.2, and 3), three AtSWEETs (AtSWEET1–3), six OsSWEETs (OsSWEET1a, 1b, 2a, 2b, 3a, and 3b) and two VvSWEETs (VvSWEET1 and 2). Two VmSWEETs (VmSWEET4 and 5), five AtSWEETs (AtSWEET4–8), nine OsSWEETs (OsSWEET7a–7e, 6a, 6b, 4, and 5) and four VvSWEETs (VvSWEET4, 5a, 5b, and 7) belonged to Clade II. Five VmSWEETs (VmSWEET10.1, 10.2, 12, 13, and 14), seven AtSWEETs (9–15), five OsSWEETs (OsSWEET11–15) and five VvSWEETs (VvSWEET9, 10, 11, 12, and 15) were included in Clade III. Clade IV had the the fewest members and contained one VmSWEET (VmSWEET16), one OsSWEETs (OsSWEET16), two AtSWEETs (AtSWEET16 and 17), and three VvSWEETs (VvSWEET17a, 17b, and 17d).

Multiple sequence alignment, conserved domain and gene structure analysis of VmSWEETs

The result of multiple sequence alignment is presented in File S7. The amino acid sequence identity among the 13 VmSWEETs ranged from 18–70%. The majority of VmSWEET members contained two MtN3/saliva domains, also known as PQ-loop-repeat, which consist of 3 + 1 + 3 transmembrane helices. Four serine (S), two tyrosine(Y) sites, and one threonine(T) phosphorylation site were predicted in the two conserved MtN3/saliva regions and were indicated by the red triangles. Additionally, to search for the key amino acid sites for VmSWEETs binding to sugars, we found a highly conserved pair of asparagine residues (N77 and N197), which were located in the binding pocket of OsSWEET2b in rice, and identified at equivalent VmSWEET positions. Furthermore, S54 on THB1 (the first MtN3 unit in a SWEET) and W76 on THB2 (the second MtN3 unit in a SWEET) have been confirmed to play analogous role in AtSWEET1 (Tao et al., 2015). In all VmSWEETs, W was present at the equivalent position of W176, except for in VmSWEET2.1, 13, and 14, where it was replaced with aromatic residue F or Y. Similarly, at the corresponding S54 position, it was replaced with F, W, L, Y, or C.

The conserved motifs were predicted to provide more insights into the characteristics of VmSWEET genes. As shown in Fig. 3B, a total of nine different conserved motifs were identified. Detailed information for each motif is provided in File S8. Motif 2, motif 4, and motif 5 were observed in all 13 VmSWEET proteins. Motif 2 and motif 4 belong to the first conserved MtN3/saliva domain, while motif 5 belongs to the second conserved MtN3/saliva domain. These findings suggest that the three conserved motifs may be essential for cranberry SWEET protein function. The conserved motifs within the N-terminus of most VmSWEET proteins exhibited the same order (motif 8, motif 4, and motif 2), except when motif 8 was absent in VmSWEET2.1, VmSWEET4, VmSWEET5, and VmSWEET16. Significantly, motif 7 was exclusively present in VmSWEET16 in Clade IV, suggesting a specific function. Motif 1 was not present in VmSWEET2.2, VmSWEET5, VmSWEET13, and VmSWEET14, but an additional transmembrane-domain structure appeared at the same position (Fig. 3C).

Figure 3 Conserved motifs and conserved structural domains of the cranberry SWEET gene family.

(A) The phylogenetic tree of VmSWEETs. (B) The conserved motifs of VmSWEET members. The colored squares correspond to nine different conserved motifs. (C) Conserved structural domains of VmSWEET genes. The green square represents MtN3-slv domain, yellow square represent s PQ-loop domain, pink square represents transmembrane- domain , blue square represents low-complexity-region. X-axis represents the number of amino acids.

To further investigate the structural differences in VmSWEET genes, the arrangement of introns and exons was determined. As shown in File S9, the number of exons in the 13 VmSWEET genes ranged from 4–9, eith the number of introns correspondingly ranging from 3–8. The number of introns in Clade I versus Clade III differed significantly, VmSWEET genes varied from 4–8 in Clade I and from 3–5 in Clade III. However, gene pairs in the sister branch exhibited similar structural features, such as VmSWEET1.1 and VmSWEET1.2, VmSWEET2.1 and VmSWEET2.2, VmSWEET10.1 and VmSWEET10.2, and VmSWEET12 and VmSWEET14, with comparable intron and exon numbers and CDS lengths. Additionally, VmSWEETs in Clades II and IV exhibited the same exon count of 4.

Chromosomal localization and duplication analysis of VmSWEETs

Gene duplication events were the main drivers of SWEET gene family expansion and these included segmental and tandem duplications. Tandem replication occurs within regions of chromosome recombination and results in the formation of gene family members that are typically closely arranged on the same chromosome, thereby constituting a gene cluster with homologous sequences and similar functionalities. However, genes arising from segmental duplication are widely separated and sometimes located on distinct chromosomes. Chromosomal localization and synteny analysis were conducted to study the repetitive events in the SWEET gene family. As shown in File S10, the 13 VmSWEET genes were unevenly distributed across the cranberry chromosomes. Chromosome 5 exhibited the highest number of genes mapped, which highlights the proximity of VmSWEET10.2 and VmSWEET12. Chromosomes 1 and 4 contained two VmSWEET genes each. Notably, the distance between VmSWEET10.1 and VmSWEET13 on chromosome 4 was remarkably short, spanning approximately 27.9 kb. Chromosomes 2, 3, 6, 9, and 12 each contained just one VmSWEET gene. Additionally, synteny relationships were analyzed to investigate potential evolutionary mechanisms in the VmSWEET gene family. The result showed synteny existed in VmSWEET14 located on chromosome 1 and VmSWEET10.1 located on chromosome 4, as well as VmSWEET4 located on chromosome 3 and VmSWEET5 located on chromosome 5, indicating two pairs of segmental duplicated events in the evolution of cranberry (Fig. 4).

Figure 4 Duplication analysis of SWEET gene family in cranberry.

Each box represents a scaffold, the number beside the box represents the position on chromosome. Gray lines indicate all synteny blocks in the cranberry genome, and the red lines indicate the duplication of VmSWEET gene pair.

Promoter cis-acting elements analysis of VmSWEET genes

To investigate the potential regulatory factors in VmSWEET genes, promoter cis-regulatory elements were predicted using PlantCARE. A total of 79 cis-acting elements were identified in the promoter regions of cranberry SWEET genes (Fig. 5). Besides the necessary components for normal transcriptional activity, such as CAAT and TATA elements, the rest were related to plant hormone, light responsive, growth and development, and stress responses. The growth and development responsive elements included meristem expression (CAT-box), HD-Zip1/HD-Zip3 (differentiation of palisade mesophyll cells), MSA-like (cycle regulation), and RY-element (seed-specific regulation). The stress responsive elements included ARE (anaerobic induction responsive element), MBS/MYC (drought stress responsive element), LTR (low-temperature responsive element), WUN-motif (wound-responsive element), and MYB/TC-rich repeats (defense and stress responsive elements). The hormone responsive elements included TCA-element/AuxRR-core (salicylic acid responsive element), TGA-element (a type of auxin responsive element), ABRE (abscisic acid responsive element), TGACG-motif/CGTCA-motif (methyl-jasmonate responsive element), GARE-motif/P-box/TATC-box (gibberellin responsive element), and ERE (ethylene responsive element). The number of light responsive elements were the least common and included G-box/GT1-motif (light responsive element) and circadian (circadian rhythm regulatory cis-acting elements).

Figure 5 Promoter cis-acting elements of VmSWEETs.

Cis-elements related to stress responses: LTR, GC-motif, ARE, WUN-motif, CCGTCC motif, DRE, DRE 1, DRE core, MBS, TC-rich repeats, box s, and AP-1. Cis-elements related to growth and development: AT-rich sequence, AT-rich element, CCGTCC-box, AE-Box, AC II, AC I, RY-element, MBSI, E2Fb, CAT-box, circadian, HD-Zip 1, HD-Zip 3, MSA-like, CCAAT box, and O2-site. Light-responsive elements: MRE, Box-4, G-box, GT1-motif, GA-motif, ATCT-motif, LAMP-element, TCT-motif, chs-CMA2a, chs-CMA1a, chs-unit1 m1, I-box, GATA-motif, Gap-box, and TCCC-motif. Cis-elements related to hormone: GARE-motif, P-box, TATC-box, ABRE 4, ABRE 3a, ABRE 2, ABRE, F-box, AuxRR-core, TGA-element, CGTCA-motif, TGACG-motif, ERE, and TCA-element. Unknown elements: AT TATA box, TATA, CTAG-motif, CARE, TCA, dOCT, as-1, A-Box, MYB recognition site, Myb-binding site, MYB, Myb, MYB-like sequence, MYC, Myc, STRE, AAGAA motif, box III, and box II. Core promoter: TATA-box and CAAT-box.

Expression profiles of VmSWEET genes in different tissues and fruit development stages of cranberry

Spatiotemporal expression patterns of 13 VmSWEET genes were determined by qRT-PCR to investigate the functions of VmSWEET genes in cranberry growth and fruit ripening. As illustrated in Fig. 6, VmSWEET4, VmSWEET5, VmSWEET10.1, VmSWEET12, and VmSWEET14 demonstrated significantly higher expression levels in flowers compared to roots, stems, and leaves. It was noteworthy that VmSWEET10.1 displayed the highest expression among these five genes. The expression of four VmSWEET genes (VmSWEET2.1, VmSWEET2.2, VmSWEET13, and VmSWEET16) was predominantly observed in the upright and runner stems. Specifically, the expression of VmSWEET16 in upright and runner stems was 6–23 fold and 10–42 fold higher than other tissues respectively. Although VmSWEET13 exhibited higher expression in upright stems compared to runner stems, no statistically significant difference was observed between the two types of stems. VmSWEET3 and VmSWEET10.2 exhibited similar expression patterns, with significantly higher expression in upright and runner leaves compared to other tissues. VmSWEET1.2 had the highest relative expression, not only in runner leaves but also in flowers. However, its expression was lower than that of VmSWEET3 and VmSWEET10.1, which were specifically expressed in flowers and runner leaves. No VmSWEET exhibited specific expression in the roots. VmSWEET16 had the highest expression among all members in the roots, but the level of expression remained extremely low.

Figure 6 Expression analysis of VmSWEET genes in different tissues of cranberry.

Rt, Roots; Ur, Upright stems; Rn, Runner stem s; UrL, Leaves of upright stem ; RnL, Leaves of runner stem ; F, Flowers. Each value is the mean of three biological replicates, and the height of the vertical bar represents the standard deviation. Different lowercase letters represent the significant statistical difference between the different groups at P < 0.05.

The expression patterns of VmSWEET genes were different at the four distinct stages of fruit development (Fig. 7). VmSWEET1.1, VmSWEET1.2, VmSWEET5, VmSWEET10.2, VmSWEET13, and VmSWEET14 exhibited similar expression profiles, which were characterized by an initial upregulation (S1–S2) followed by a subsequent downregulation during fruit development (S3–S4), with a peak value during S2 that was significantly higher than in other stages. It was notable that the expression of VmSWEET14 was the highest among all members. Compared with the fruit at S2, significant decreases in VmSWEET14 expression of 78.43% and 94.90% occurred at S3 and S4, respectively. In contrast, VmSWEET4, VmSWEET10.1, and VmSWEET12 had “high-low-high” patterns of expression. Among them, VmSWEET10.1 and VmSWEET12 exhibited weak expression throughout S2. VmSWEET4 exhibited a significant decrease in expression from S1–S2, followed by a significant increase upon S4, but no significant differences were observed between the late stages. Additionally, the expression of VmSWEET2.1, VmSWEET2.2, VmSWEET3, and VmSWEET16 gradually declined during fruit development. Compared with the fruit at S1, VmSWEET16, with the highest expression level among the four genes, experienced significant reductions of 46.96%, 91.94%, and 94.45% in the fruit at S2, S3 and S4, respectively.

Figure 7 Expression analysis of VmSWEET genes in cranberry fruits at different developmental stages.

The X-axis labels indicate cranberry fruits at different developmental stages. S1, Young fruit stage; S2, Fruit expansion stage; S3, Color turning stage; S4, Maturity stage. Different lowercase letters represent the significant statistical difference between the different groups at P < 0:05.

Expression profile of VmSWEET genes in response to abiotic stress

In vitro cranberry plantlets received abiotic stress treatments (osmotic, saline, saline-alkaline and aluminum) to investigate the differential expression patterns of VmSWEET genes (Fig. 8). Under osmotic conditions, the most prominent finding was that VmSWEET10.2 and VmSWEET14 exhibited the highest expression levels in the SWEET gene family, while displaying contrasting trends. VmSWEET10.2 increased sharply within the first 9 h and subsequently decreased, and significant differences were observed among different treatments, except for 12 h and 24 h. Conversely, the expression of VmSWEET14, which was initially low, decreased within the first 6 h and subsequently significantly upregulated until reaching peak value at 24 h. Notably, peak expression was 14.5-fold higher than that of the control. Other genes exhibited relatively low expression and little fluctuation. For instance, the expression levels of VmSWEET1.2, VmSWEET2.1, VmSWEET10.1, and VmSWEET12 were downregulated, whereas VmSWEET5, VmSWEET2.2, and VmSWEET13 were upregulated. Under the saline stress, VmSWEET10.2 and VmSWEET14 exhibited high expression with an upward trend over time. However, their response times differed. A significant upregulation of VmSWEET10.2 was observed within the first 3 h, after which there was no significant difference in expression at 6, 9, and 12 h compared with that at 3 h. However, the expression of VmSWEET14 did not exhibit a significant increase until 24 h, at which time it was 13.5 times higher than that of the control. During the saline-alkaline treatment, VmSWEET10.2 consistently exhibited the highest expression among all genes and displayed significant upregulation, with expression progressively increasing 5.9, 6.6, 8.2, 10.1, and 11.0-fold over time. Furthermore, significant differences were observed between different treatments. In response to aluminum stress, VmSWEET10.2 exhibited slightly increased expression within the first 9 h, and then peaked at 12 h with an expression level 20-fold higher than that of the control. Subsequently, a significant decrease was observed. Other genes were expressed at low levels and were continuously downregulated under aluminum stress, and these genes included VmSWEET1.2, VmSWEET2.1, VmSWEET2.2, VmSWEET4, VmSWEET5, VmSWEET12, and VmSWEET16.

Figure 8 Gene expression heatmap of the VmSWEET genes in cranberry leaves under various abiotic stresses.

The X-axis labels indicate the time points at which samples were collected (0, 3, 6, 9, 12, and 24 h) during the various stress treatments. Red and blue correspond to strong and weak expression of the VmSWEE genes, respectively.

Discussion

Characters and function of SWEET family genes in cranberry

The SWEET gene family is widely present in plants, animals, fungi, and bacteria, and these transporters mediate bidirectional cross-membrane movement of sugars through an alternating access mechanism to regulate various life activities (Eom, Chen & Sosso, 2015; Latorraca et al., 2017). The SWEET gene family has been extensively characterized in many plant species due to the advanced of high-throughput sequencing techniques, such as genomic and transcriptome sequencing. In general, plant genome ananlyses have revealed an approximate presence of 20 SWEET paralogs (Anjali et al., 2020). In this study, 13 SWEET genes were identified in cranberry via a comprehensive genome-wide investigation. The number of VmSWEETs is comparable to the number in tea (13) (Wang et al., 2018), but less than the number in soybean (52) (Patil et al., 2015), oilseed rape (68) (Jian et al., 2016), or wheat (108) (Gautam et al., 2019). Variation in gene family size between different species can be attributed to gene duplication events, which play a vital role in the evolution of gene families (Yin et al., 2013). In rice, the monosaccharide transporter gene family exhibits a significant large size, comprising 65 genes, with two subfamilies experiencing substantial expansion through tandem duplications (Johnson & Thomas, 2007). Recent studies have revealed that the SWEET gene family underwent significant expansion during its evolution history (Patil et al., 2015). A tandem duplication event is defined as the presence of two or more genes within a chromosomal region of 200 kb (Holub, 2001), so we speculate that tandem duplications are implicated in the presence of VmSWEET10.1 and VmSWEET13 on chromosomes 4, as well as VmSWEET10.2 and VmSWEET12 on chromosomes 5. Descendant genes, as a result of segmental duplications, are far apart and even located on different chromosomes. According to synteny analysis, VmSWEET gene duplication involved two segmental duplications on chromosomes1, 4, 3, and 5 (VmSWEET4/VmSWEET5 and VmSWEET10.1/VmSWEET14), which is less than large-scale gene duplication events reported in soybeans (Schmutz et al., 2010). Segmental duplication is prevalent in plants due to their diploidized polyploid nature, which results in the retention of multiple duplicated chromosomal blocks within genomes (Cannon et al., 2004). Cranberry is a diploid plant, in contrast to other higher plants that have undergone genome duplication, such as wheat, oilseed rape, and soybeans, the lack of a diploidized polyploid may be the primary reason for the limited number of VmSWEETs.

Based on the evolutionary relationships inferred from phylogenetic analysis, the VmSWEET genes were categorized into four distinct clades (Fig. 2), that were determined by SWEET preferences for monosaccharides or disaccharides. Those in Clades I and II specifically transport hexose, Clade III members display preferentially transport sucrose over glucose, and Clade IV members specifically transport fructose (Eom, Chen & Sosso, 2015). According to their putative subcellular localizations and assigned clades, we speculate that tonoplast-localized VmSWEET1.1 and VmSWEET2.2 facilitated transmembrane transport of hexoses, such as glucose and fructose, while VmSWEET2.1, VmSWEET3, VmSWEET4, and VmSWEET5 facilitate the transport of hexoses across the plasma membrane. VmSWEET10.1, VmSWEET10.2, VmSWEET12, VmSWEET13, and VmSWEET14, putatively located on the plasma membrane, may efflux sucrose from the cytosol into the apoplast. Additionally, VmSWEET16 in Clade IV may control the flux of fructose across the plasma membrane. The precise subcellular localization and substrate specificity of VmSWEETs requires further research. To understand substrate specificity of SWEETs, crystal structure and bioinformatic analyses were conducted in bacterial SemiSWEETs (Wang et al., 2014). An interesting finding was that the size of the pocket presented above the center of the transporter protein played a critical role in determining substrate specificity. A larger substrate-binding pocket with a spacious substrate-binding cavity may facilitate the transport of disaccharides (such as sucrose) and monosaccharides (such as glucose and fructose), while smaller pockets with a restricted substrate-binding cavity can only hold monosaccharides (Wang et al., 2014). In higher plants, a conserved asparagine pair (N77 and N197) surrounds the binding pocket at the equivalent positions in OsSWEET2b, and S54 on THB1 and W176 on THB2 have also been implicated in the transportation capacity of AtSWEET1 (Tao et al., 2015). In our study, N77 was conserved in all VmSWEETs, while the conservation of N197 was observed in the majority of VmSWEETs, except for an E substitution at the equivalent positions of VmSWEET13 and VmSWEET14. In addition, most VmSWEETs contained W at the positions equivalent to W176, but exceptions were observed in VmSWEET2.1, 13, and 14, where W was substituted with aromatic residue F or Y (File S7). Nevertheless, this substitution may not affect the transport activity of VmSWEETs, because the presence of one aromatic residue in THB2, rather than THB1, was important for transport activity (Tao et al., 2015). We speculate that four amino acid residues of VmSWEET can still interact with sugar molecules via H-bonding or aromatic ring stacking. Phosphorylation sites were also crucial for protein function and interaction. Lately, research has shown that the carboxy-cytosolic regions of AtSWEET11 and 12 were rapidly phosphorylated by SnRK2 protein kinases upon drought, which enhances the oligomerization and sucrose transport activity of SWEETs (Fatima, Anjali & Senthil-Kumar, 2022). In our study, four S, two Y and one T phosphorylation site were observed in the conserved domains of VmSWEETs (File S7). These conserved phosphorylation sites were also identified within CsSWEETs of watermelon (Xuan et al., 2021). Combined with the qPCR results, we suggest that the seven phosphorylation sites are probably related to signal recognition and transduction functions of VmSWEETs that enable responses to multiple types of stress.

The SWEET gene family in plants is highly conserved, with accurate functioning and stability maintained by seven TMDs and two MtN3/saliva domains (Chen et al., 2010). Conserved structural domains analysis revealed that 11 VmSWEET proteins (about 85%) contained two complete MtN3/saliva domains, while VmSWEET2.2, VmSWEET13, and VmSWEET14 only exhibited one MtN3/saliva domain with 5–6 TMDs (Fig. 3C and Table 1). Despite possessing two MtN3/saliva domains, both VmSWEET5 and VmSWEET6 exhibited a discrepancy in the number of transmembrane helices predicted by use of the TMHMM 2.0 Tool, with only six instead of the expected seven TMDs identified in this analysis. Similar observations of SWEET members containing one or one and a half MtN3/saliva domains have been reported in other species, such as walnut (Jiang et al., 2020) and watermelon (Xuan et al., 2021). The presence of SWEET proteins containing two MtN3/saliva domains in eukaryotes has been attributed to replication or horizontal gene transfer from the one MtN3/saliva domain of prokaryotes (Xuan et al., 2013). Therefore, we hypothesized that incomplete VmSWEET genes, VmSWEET2.2, VmSWEET13, and VmSWEET14, were generated through tandem and domain duplication events during the course of evolution. Phylogenetic analyses also supported the results of gene structure analysis. Minimal variation in the number of introns and exons was observed within each clade, except for VmSWE ET3 in Clade I. The gene pairs in the sister branches generally had the same number of introns and exons, suggesting that molecular features of SWEET genes were conserved during evolution. Introns serve as hallmark features of eukaryotic genes and contribute to genetic diversity through alternative splicing (Jeena, Kumar & Shukla, 2019). There is a difference in the number of introns between unicellular and multicellular organisms. For example, fungi or oomycetes have no or few introns, while plants contain 4–5 introns per gene (Hu et al., 2016). In cranberry, VmSWEETs contain 3–8 introns, which is similar to tomato (Feng et al., 2015) and watermelon (Xuan et al., 2021). Conserved motif analyses revealed the presence of motif 2, motif 4 and motif 5 in all VmSWEET proteins. Motif 2 and motif 4 belong to THB1, while motif 5 belongs to THB2, suggesting their crucial role in maintaining structure integrity and functional efficacy. Additionally, gene members within the same clade exhibited similar motif arrangement, while there were obvious differences in the motif composition among different clades. For instance, motif 7 was uniquely present in members of Clade IV and motif 8 was specifically present in members of Clade I and III. These specific motifs were not found in members of the remaining two clades (Fig. 3). These results were consistent with other plant systems, such as rice (Yuan & Wang, 2013), banana (Miao et al., 2017) and wheat (Gautam et al., 2019).

Gene expression and functional divergence of SWEETs in Cranberry

The expression profile of a gene is closely related to its function. Previous studies revealed the importance SWEETs in plant growth and development. In this study, the expression patterns of 13 VmSWEET genes were analyzed in roots, stems, leaves, flowers and different development stages of fruit to explore the potential function of SWEET genes in cranberry. The results revealed distinct expression patterns for VmSWEET genes in different tissues (Fig. 6). Notably, VmSWEET3 and VmSWEET10.2 were highly expressed in leaves of upright and runner stems. VvSWEET1, the homolog of VmSWEET3, was previously shown to be expressed in young and mature leaves of grape (Chong et al., 2014), and to have an expression pattern in vegetative organs that was similar to VmSWEET7. AtSWEET11 and AtSWEET12, which were clustered in Clade III with VmSWEET2, were highly expressed in leaves and played crucial roles in sugar efflux from mesophyll cells to the apoplast in Arabidopsis, while the atsweet11;12 mutant lines were smaller and accumulated starch in leaves (Chen et al., 2012). Because sucrose is the predominant photoassimilate that is transported in Clade III, it has been hypothesized that VmSWEET10.2 plays a role in the phloem loading of photoassimilates in cranberry leaves. The SWEET genes expressed in flowers primarily participate in reproductive development and nectar secretion (Eom, Chen & Sosso, 2015; Wen et al., 2022; Wen et al., 2022; Lin et al., 2014). VmSWEET10.1 exhibited the highest transcriptional level in flowers. Phylogenetic analysis showed that OsSWEET11, AtSWEET13,14, and VmSWEET10.1 belonged to the same clade. Among them, OsSWEET11 has been reported to play a role in rice pollen development, ossweet11 knockouts produced defective pollen grains and had a lower fertility rate (Chu et al., 2006; Yang, Sugio & White, 2006; Yuan et al., 2009). Consistent results were reported in Arabidopsis, AtSWEET13 and AtSWEET14 were expressed in the anther wall, which facilitated sucrose efflux into locules to support pollen development and maturation. Consequently, an atsweet13;14 mutant displayed decreased viability and germination of pollen (Sun et al., 2013; Wang et al., 2022). Therefore, VmSWEET10.1 may play an important role in cranberry reproductive development. From source to sinks, the long-distance transportation of photosynthetic products in stems generally follows the symplastic route. However, when stems function as storage organs, SWEETs may be involved in unloading and storage of photosynthates in the stem. For example, SsSWEET4a/4b were mainly expressed in the stems of sugarcane and were forecasted to be involved in sugar transportation within the stalk (Hu et al., 2018). Although the stems of cranberry do not serve as storage sinks like those in sugarcane, the expression of VmSWEET16 in upright stems and runner stems was higher than other tissues. To understand the role of SWEETs in plant stems, further functional validation is required. No VmSWEET genes were specifically expressed in the roots, because roots might not serve as an important storage sink during the sampling period.

Fruits are the most important storage organs in horticultural crops, their yield and quality are determined by the content of sugar. As a novel sugar transporter that function independently of energy or pH, SWEET proteins have attracted attention in the context of phloem unloading, transport and storage of sugars during fruit development. In jujube, the expression of ZjSWEET11 and ZjSWEET18 gradually increased during fruit development, peaking at complete maturity (Yang et al., 2023). In apple, there was a significant association between the expression of MdSWEET2e, 9b, 15 and fruit sugar content. In particular, MdSWEET15a and MdSWEET9b accounted for a large proportion of phenotypic variation in sugar content (Zhen et al., 2018). In grape, VvSWEET10 was strongly expressed in ripening fruit, and VvSWEET10 overexpression in grapevine calli and tomatoes resulted in a significant increase in glucose, fructose and total sugar (Zhang et al., 2019b). In developing tomato fruits, SlSWEET15 expression was notably elevated, while fruit sizes and weights were significantly reduced upon elimination of SlSWEET15 (Ko et al., 2021). Together, the above results all indicate that SWEET genes exert a positive regulatory effect on fruit development and ripening. Conversely, silencing SlSWEET7a or SlSWEET14 in tomato increased plant height, fruit size, and sugar content (Zhang et al., 2021). In our study, expression of VmSWEETs changed dynamically during fruit development, with distinct sets of VmSWEETs being expressed in the young and mature fruits (Fig. 7). For instance, VmSWEET14 and VmSWEET16 were highly expressed during the young fruit stage and expansion stage, respectively, whereas VmSWEET4 was highly expressed during the color change and the maturity stage. We speculate that VmSWEET4 positively regulates fruit development and ripening, while VmSWEET14 and VmSWEET16 may play roles that are similar to with SlSWEET7a and SlSWEET14. Suppressing the two genes could be a potential strategy for enhancing the sugar content of cranberry fruit.

Abiotic stress frequently impedes plant growth and development, and ultimately inhibits plant productivity and quality. Damage to sucrose phloem transport and source/sink relationships is an important factor (Lemoine et al., 2013). Plants have evolved sensory and response mechanisms to cope with various environmental stresses. Sugars serve as osmo-protectants and molecular switches, and their production and distribution are crucial physiological processes that are induced by various stresses (Saddhe, Manuka & Penna, 2021). Previous studies found that SWEET proteins can regulate the redistribution of soluble sugars under abiotic stress, but expression patterns differed. For example, in bluegrass (Zhang, Niu & Ma, 2020), cotton (GhSWEET5, 20, 49, and 50) (Li et al., 2018), tea (CsWEET1a, 2a, 2c, 3a, 7a, 7b, and 10) (Jiang et al., 2021), and wheat (TaSWEET14g-1A and 16a-4A) (Gautam et al., 2019) SWEET genes were induced by drought or osmotic stress, while MtSWEET2a and MtSWEET3c were downregulated in Medicago truncatula (Hu et al., 2019). In this research, the most noticeable result was that the expression of VmSWEET10.2 was the highest of all the VmSWEET s, and it was upregulated under all abiotic stress treatments (Fig. 8). The expression patterns in osmotic and saline treatments were consistent with those in the homologs AtSWEET11, 12, 14, and 15, which have been demonstrated to respond to a variety of abiotic stresses in Arabidopsis (Durand et al., 2016; Sellami et al., 2019; Seo et al., 2011). AtSWEET11 and AtSWEET12 were upregulated and were responsible for the transport of sucrose from the leaves to the roots in water deficit plants (Durand et al., 2016). AtSWEET14 was upregulated in response to high salinity (Sellami et al., 2019), and AtSWEET1 5 (also known as SAG29) was significantly upregulated during senescence and abiotic stresses that included cold, salinity, and drought (Seo et al., 2011). In our study, VmSWEET10.2 also responded to saline-alkaline and aluminum treatments, suggesting its potential roles in regulating sucrose transport and distribution under abiotic stress. Previous research demonstrated that drought and salinity induced an ABA-responsive transcription factor OsbZIP72 to bind directly to the promoters of OsSWEET13 and 15, thereby activating their transcription and increasing the sucrose content in leaves and roots (Mathan, Singh & Ranjan, 2020). Thus, we predicted that VmSWEET10.2 might harbor a site for the ABA-responsive transcription factor in its promoter region, similar to the homologues of OsSWEET13 and 15. This conjecture was consistent with the presence of ABRE (abscisic acid response element) by promoter analysis, but the regulatory mechanism of sugar homeostasis in cranberry under abiotic stress requires further exploration.

According to the expression patterns of VmSWEET genes in different tissues and at different fruit development stages, we propose a hypothetical model for SWEETs involved in the transport and distribution of photosynthetic products in cranberry. As illustrated in Fig. 9, sucrose is produced in upright and runner leaves through photosynthesis, and VmSWEET3 participates in phloem loading of photosynthetic products in both types of leaves. Then, long-distance transport of sucrose from source to sink tissues in upright stems and runner stems is facilitated by VmSWEET16. V mSWEET10.1 is likely to be implicated in pollen development in flowers, which benefits pollination and fertilization. With sucrose unloading into the fruit, VmSWEET14 and VmSWEET16 play an important role in the early stage of fruit growth and development, while VmSWEET4 is responsible for the transport and accumulation of monosaccharides (hexoses) during the S2 and S3. VmSWEET10.2 may be induced by abiotic stress to transport sucrose in roots as a signaling molecule to cope with different constraints.

Figure 9 Schematic model of preferential gene expression and proposed roles of VmSWEETs in different cranberry tissues and fruit development stages.

This figure shows the representative genes highly expressed in each tissue and fruit development stage during the sugar accumulation stage, i.e., those likely implicated in the process of sucrose transportation from the leaf to other plant organs, such as the flower, stem, and fruit. The gene names under the tissue’s name indicate that they are highly expressed in those tissues.

Conclusion

In recent years, numerous important advances in the study of SWEET transporters have been reported, but several unresolved issues persist. For instance, it remains unclear whether the members of the SWEET gene family in plants function independently or collectively. Additionally, the relationship between structure and function requires further exploration. Questions remain regarding how SWEET proteins are regulated and whether they are regulated at the transcriptional or translational level. In this study, 13 VmSWEET genes were identified in cranberry, they were classified into four clades and distributed on eight chromosomes. Four conserved conserved amino acid residues and seven phosphorylation sites, which might be crucial for transport, were observed in the conserved domains. Cis-acting elements of VmSWEET promoters were related to plant hormone, light, growth, development, and stress responses. The expression of VmSWEETs varied across different tissues and fruit developmental stages. VmSWEET3, VmSWEET16, and VmSWEET10.1 were specifically expressed in leaves, stems, and flowers, respectively. VmSWEET4, VmSWEET14, and VmSWEET16 might play crucial roles in fruit development and ripening. VmSWEET1.02 might be the key gene involved in the response of cranberry to abiotic stress during osmotic, saline, saline-alkaline and aluminum treatments. These results provide a foundation for future studies of VmSWEET gene function and provide a basis for improving yield, quality, and resistance in cranberry plants.

Supplemental Information

Supplemental Information 1 CDS sequences of VmSWEET genes in cranberry

Supplemental Information 2 Promoter sequences and cis-acting elements of VmSWEET genes in cranberry

Supplemental Information 3 The amino acid sequences used to phylogenetic analyses and multiple sequence alignment

Supplemental Information 4 qRT-PCR primers of VmSWEET genes in cranberry

Supplemental Information 5 The raw data of Ct value used for qRT-PCR

Supplemental Information 6 MIQE checklist

Supplemental Information 7 Multiple sequence alignment of OsSWEET2b, AtSWEET1 and VmSWEETs

The sequences contained in the black boxes are conserved domains of VmSWEETs members. The position of the S, T, and Y predicted to be the phosphorylation sites are indicated by the red triangles. A conserved asparagine pair Asparagine (N77 and N197) in OsSWEET2b and a serine (S54) as well as tryptophan (W176) in AtSWEET1 , which surround the binding pocket associated with the transportation capacity, are indicated by the red arrows.

Supplemental Information 8 The different conserved motifs in cranberry

The letter size of the amino acid represents the frequency of the corresponding nucleotide.

Supplemental Information 9 Gene structure of cranberry SWEET gene family

Supplemental Information 10 Chromosome mapping of SWEET genes in cranberry

Additional Information and Declarations

Competing Interests

Author Contributions

DNA Deposition

Data Availability

The authors declare there are no competing interests.

Li Chen conceived and designed the experiments, performed the experiments, prepared figures and/or tables, authored or reviewed drafts of the article, and approved the final draft.

Mingyu Cai performed the experiments, analyzed the data, prepared figures and/or tables, authored or reviewed drafts of the article, and approved the final draft.

Jiaxin Liu performed the experiments, analyzed the data, prepared figures and/or tables, authored or reviewed drafts of the article, and approved the final draft.

Xuxin Jiang performed the experiments, analyzed the data, prepared figures and/or tables, authored or reviewed drafts of the article, and approved the final draft.

Jiayi Liu performed the experiments, analyzed the data, prepared figures and/or tables, authored or reviewed drafts of the article, and approved the final draft.

Wang Zhenxing conceived and designed the experiments, prepared figures and/or tables, authored or reviewed drafts of the article, and approved the final draft.

Yunpeng Wang conceived and designed the experiments, performed the experiments, analyzed the data, prepared figures and/or tables, authored or reviewed drafts of the article, and approved the final draft.

Yadong Li conceived and designed the experiments, authored or reviewed drafts of the article, and approved the final draft.

The following information was supplied regarding the deposition of DNA sequences:

The data are available at GenBank: KJ826367 to KJ826395.

The following information was supplied regarding data availability:

The data is available at NCBI GEO: ASM2260669v1.

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
