# Peer review of "Genome-wide identification and expression analyses of SWEET gene family reveal potential roles in plant development, fruit ripening and abiotic stress responses in cranberry (Vaccinium macrocarpon Ait)"

_PeerJ, doi:10.7717/peerj.17974_

## Round 0.1 · original submission · Major Revisions

Dear authors,

Reviewers #2 and #3 have identified extensive lists of issues with your submission to PeerJ.

Both peer reviewers have provided annotated PDFs post their review of your study - please use these together the the Reviewer-specific comments, to prepare a significantly improved manuscript to PeerJ for subsequent re-consideration of its further merits.

All comments and concerns raised by reviewers #2 and #3 must be addressed as part of preparing a significantly improved manuscript version. Failure to adequately address the concerns/comments by the two expert reviewers will result in the resubmitted manuscript not being further considered for publication in PeerJ.

Considering that 'Major Revision' forms the decision at this time, please take your time when preparing the revised version of your study.

All the very best with preparing the revised manuscript version.
Kind regards,
Andrew Eamens

**Language Note:** PeerJ staff have identified that the English language needs to be improved. When you prepare your next revision, please either (i) have a colleague who is proficient in English and familiar with the subject matter review your manuscript, or (ii) contact a professional editing service to review your manuscript. PeerJ can provide language editing services - you can contact us at [email protected] for pricing (be sure to provide your manuscript number and title). – PeerJ Staff

Reviewer 1 ·

Basic reporting

The results are well presented and the data presented in the figures and tables are easy to understand and well presented. The English is clear and easy to understand, however, there are some small mistakes throughout and these should be corrected here are some examples from the introduction
Ln 47 involves not involve
Ln 49 should be 'the apoplastic'
ln 50 requires not require
ln 51 key redundant word. transporter not transporters
ln 56 bi-directions is bad grammar
ln 71 devoted not a good word. Discovery of SWEET gene no good better to say the first SWEET gene
ln 73/74 actually it was known since the work of Giaquinta et al in the late 70's that sucrose was passively unloaded from the mesophyll into the apoplast and actively loaded into the phloem. The discovery of SWEET identified the transporter involved in unloading from the mesophyll
ln 85 the extracellular
Ln 94 The above
Ln 95 the unloading
Figure 1 legend - makes no sense. Should be - Different cranberry tissues and fruit at different stages of development
The authors' should check through the entire manuscipts for similar mistakes and correct them
An informative and comprehensive introduction is provided and the discussion is appropriate.

Experimental design

no comment

Validity of the findings

no comment

Additional comments

This paper provides a thorough characterisation of SWEET's in cranberry.

My view is that the paper provides new and important information, and is suitable for publication after minor editing as mentioned above

·

Basic reporting

1. Basic Reporting
Clear and unambiguous, professional English used throughout:

-Reviewer comment 1 : Basic English redaction mistakes and grammatical errors are present throughout the manuscript.
Ex1 in the abstract : was highly expressed in “uprights and runners stem”, instead of “upright and runner stems”.
Ex2 in the abstract : “While VmSWEET10 expressed higher in color transition and maturity stages than in early development stages”. - This sentence seems incorrect.
Ex3 in the abstract : “Overall, these results provide new insights into the characteristics and the evolution of VmSWEET genes, and the important candidate VmSWEET genes involved in the growth and development as well as abiotic stress responses in cranberry can be explored for promoting molecular breeding to improve fruit quality and abiotic stress resistance.” - These sentence is too long and contained basic English mistakes.
These are just three examples (taken from the abstract), but many other mistakes are present in the manuscript. Please professionally review the entire manuscript accordingly, before further re-submission.


Literature references, sufficient field background/context provided.

-Reviewer Comment 2 : I found that some references were missing. Please at least introduce or discuss the following references:
1- Lemoine R, La Camera S, Atanassova R, Dédaldéchamp F, Allario T, Pourtau N, et al. Source-to-sink transport of sugar and regulation by environmental factors. Frontiers in Plant Science. 2013;4(272).
2 - Jeena GS, Kumar S, Shukla RK. Structure, evolution and diverse physiological roles of SWEET sugar transporters in plants. Plant molecular biology. 2019.
3- Johnson DA, Thomas MA. The monosaccharide transporter gene family in Arabidopsis and rice: A history of duplications, adaptive evolution, and functional divergence. Molecular Biology and Evolution. 2007;24(11):2412-23.


The article should include sufficient introduction and background to demonstrate how the work fits into the broader field of knowledge. Relevant prior literature should be appropriately referenced.

-Reviewer Comment 3: The introduction includes sufficient introduction and background. However, I would remove this part introducing TF regulating SWEET. As this manuscript does particularly focus on TF :
“Transcription factors are pivotal regulatory proteins that modulate the transcriptional rate of target genes by selectively binding to cis-acting elements of promoter upon activation or deactivation of upstream signaling cascades (Riaño-Pachón et al., 2007). Some studies have reported DNA binding with one finger (DOF) transcription factors and WRKY transcription factor can bind the promoter regions of SWEET. For example, OsDOF11 directly binds the promoter regions of OsSWEET11 and OsSWEET14 to transport sucrose via apoplastic loading (Wu et al., 2018). PuWRKY31 with high histone acetylation level directly binds to PuSWEET15 promoter then actives sucrose transporter transcription, resulting in high levels of sucrose in pear fruits (Li et al., 2020).”

-Professional article structure, figures, tables. Raw data shared :

-Reviewer comment 4: I suggest rearranging the figures, moving some info to the supplementary material:
Figure 3 may be moved to the supplemental material section. Most international articles on SWEET gene identification provide protein alignments as supplementary info. I suggest adding the phosphorylation sites in Table 1 (as a new column) if the authors want to stress on this information as main data.
Figure 5 may be moved to the supplemental material section.
Figure 6 may be moved to the supplemental material section. Most international articles on gene identification such info as supplementary materials.

-Reviewer comment 5: Please provide a vectorized image of figure 8. The current picture is pixelized and it is hard to read/ review this figure.

-Reviewer comment 6: line 448 “For instance, motif 7 was uniquely present in members of cluster IV and motif 8 was specifically present in members of cluster I and III, these specific motifs were not available in members of the remaining two clusters (Figure 4).” - Please clearly indicate clusters (I/II/II/IV) in figure 4.

-Reviewer comment 7: line560-574 - The conclusion paragraph is almost identical to the Abstract. It does not confer with PeerJ instructions. Please refer to PeerJ author guidelines : The conclusion section should contain this points : Identify unresolved questions / gaps / future directions. Please review the conclusion paragraph accordingly.


-Self-contained with relevant results to hypotheses:
Yes

Experimental design

2. Experimental design
-Original primary research within Aims and Scope of the journal.
-Yes

-Research question well defined, relevant & meaningful. It is stated how research fills an identified knowledge gap:
Yes

-Rigorous investigation performed to a high technical & ethical standard.
Reviewer comment 8 : The author used an incorrect nomenclature for naming SWEET genes. Indeed, it is not clear how the 13 VmSWEET genes were named. For instance, the single gene belonging to clade IV and orthologs of VvSWEET17a/b ant AtSWEET16/17 was named VmSWEET5. This does not make sense as SWEET5 are supposed to belong to clade II (see AtSWEET5 and OsWEET 5 for example). Thus, most VmSWEET genes were named incorrectly in the manuscript (renaming homologous VmSWEET genes could corroborate with further results on SWEET collinearity). You may refer to Doidy et al 2019 (doi 10.1371/journal.pone.0223173) for correct nomenclature on naming sugar transporter genes.
Rename all SWEET according to phylogenetic proximity with other SWEET gene from Arabidopsis, rice and grapevine. Please rename all SWEET accordingly throughout the entire manuscript, figures, tables etc...

Reviewer comment 9 : Here the authors used PEG treatment (20% PEG 8000) and named this condition as “Drought treatment”. Drougt treatment refers to the removal of water in the soil. For instance, drought treatment is comparing a plant irrigated with 100% field capacity (control condition) vs plants stressed with 50% field capacity (drought condition). Please note that PEG treatment is not considered as a “drougtht treatment” (ie. a water deficit), but is rather considered as an “osmotic stress”.
Please modify the term “drought” or any related terms in your manuscript by “osmotic stress” (if using PEG).

Reviewer comment 10:
line 425 - VmSWEETs is referring to protein, so please do not use italic font.
line 477 “double mutant AtSWEET13,14” should be written in small letters (see Wang 2022 referring to atsweet13;14 mutant).
Please review the use of italics when referring to genes, non-italics when referring to proteins, and lower-case letters when referring to mutant in the entire manuscript.


-Methods described with sufficient detail & information to replicate:

Reviewers Comment 11: Line 149 : “A typical cultivar Bain 11 planted in the small berry germplasm resource garden of Jilin Agricultural University” - is not enough to describe plant growth conditions. Please provide further information on culture conditions (soil, light, photoperiod, meteorological parmaeters, etc...).

Reviewers Comment 12: Why using two different varieties Bain11 (for developmental studies) and Bain13 (for stress studies) is not explained or discussed?

Reviewers Comment 13: line 159 : “leaf samples were collected” - At which developmental stage(s) ?

Reviewers Comment 14: line 159 “ three independent replicates” - Please add further detail whether this was 3 biological replicates (= 3 individual plants) ?

Reviewers Comment 15: line 183 “to construct an unrooted phylogenetic tree” - Please name which phylogenetic method was used to build the tree (ex neighbor joining / maximum likelihood / Parsimony …) ?

Reviewers Comment 16: line 192 “MapChart was used to construct the chromosomal distribution map of VmSWEET genes, as well as MCScanX and CRCOS were used to analysis gene synteny.” - More information is needed to understand how the synteny analysis was performed.

Validity of the findings

3. Validity of the Findings
-Impact and novelty not assessed. Meaningful replication encouraged where rationale & benefit to literature is clearly stated:
Yes


All underlying data have been provided; they are robust, statistically sound, & controlled:

-Reviewer Comment 17: -Statitstical tests performed in Figure 9 and 10 are not explained in the manuscript.

-Reviewer Comment 18 : The expression results on developmental stages (described line 304-334) and abiotic stresses (line 336-360) in the Result section of the manuscript should also use/comment the statistical tests.


-Conclusions are well stated, linked to original research question & limited to supporting results:

-Reviewer Comment 19: “line 370 The number of VmSWEETs is comparable to that of …” - Here it would be interesting to compare the number of SWEET in cranberry with other species closely related to cranberry (instead of random species).

-Reviewer Comment 20: line 377 Discussion on gene duplication - The authors did not mention/discuss the probable tandem duplications that occurred in sugar transport family evolution. For instance, refer to Johnson & Thomas 2007 (doi:10.1093/molbev/msm184).

-Reviewer Comment 21: line 440 “Generally, the gene with the highest number of introns is regarded as the ancestral homolog of those members with fewer introns, as intron loss occurs more rapidly than gain after segmental duplication (Nuruzzaman et al., 2010).” - Nuruzzaman et al., 2010 studied the NAC TF family, please make sure that your statement on intron numbers /evolution is also true for SWEET ?

-Reviewer comment 22: line 443 “VmSWEET7 with 8 introns was proposed the original homologs in cranberry SWEET gene family (Figure 5).” What does “the original homologs” mean ?

-Reviewer comment 23: Line 487 “the long-distance transportation mechanism” - Please note that long-distance transport of sugars in plants is primarily mediated by SUT proteins.

-Reviewer comment 24: line 488 “No VmSWEETT genes specifically expressed in the roots, because the root might not serve as an important storage sink during the sampling period, or SWEET transcription in root was induce by certain factors, such as cold stress, or osmotic stress.” - Please remove this statement that seems misleading.

-Reviewer comment 25: Line 535 “However, whether SWEET transporters play a role in other stress except for cold, drought and salt stress in plants remains unknown.” - This sentence seems misleading. Does the author talk about VmSWEET ? Please review or delete

-Reviewer comment 26: Line 539 “relatively autonomous.” - What does this mean ?

Additional comments

4. General Comments

-Reviewer Comment 27: Please change the term “subfamilies” and “subgroup” to “clades”. The term “clade” is clearly stated in Figure 1. However, the term “subfamily” or “subgroup” are used. Please always use the term “clade” as originally named by Chen et al 2010.

-Reviewer Comment 28: Please mention the term "cis" regulatory elements in the abstract.

-Reviewer Comment 29: Please also review all typos, mistakes etc.. highlighted on the annotated pdf document.

Reviewer 3 ·

Basic reporting

This manuscript needs extensive editing for proper English grammar. I have made many corrections in the attached, but it is likely that additional corrections are required.

The literature review is exhaustive and could probably be reduced. The Discussion should focus on cranberry vs. published work.

Figures are often mislabeled and labels don' always agree with the legends and/or text. Legends are inadequate. Figures lack sufficient labeling.

No statistical analyses are presented, but the authors frequently us the word 'significant'.

There is way too much outright speculation. No data are presented to suggest function, binding, etc.

Experimental design

This paper has essentially two parts. The first part is mainly an exercise in bioinformatics to 'find' the genes in the cranberry and make predictions, based on the reported nucleic acid and deduced amino acid sequences. Comparisons are made to published genes/proteins in the SWEET family.

The second part examines expression patterns in vegetative and reproductive tissues using RT-qPCR. While the basis work is fine, the manuscript as written suffers from the shortcomings noted above and in the attached review.

Validity of the findings

Statistical analyses should be better described.

Additional comments

Please see attached.

Annotated reviews are not available for download in order to protect the identity of reviewers who chose to remain anonymous.

---

## Round 0.2 · Minor Revisions

Dear authors,

Reviewers #2 and #3 have raised additional concerns / comments regarding your resubmission which underwent Major Revision.

Please now address the additional concerns/comments made by Reviewers #2 and #3 as part of performing the requested Minor Revisions.

I look forward to receiving your revised manuscript in the near future.

Good luck with the revisions.

All the best,
Andy

Reviewer 1 ·

Basic reporting

no comment

Experimental design

no comment

Validity of the findings

no comment

Additional comments

The authors' have carried out my revisions. My view is that the manuscript is now suitable for publication
My affiliation is not the University of Melbourne. I was formerly Universities of Perugia and Sheffield. My profile is on Research Gate Robert P Walker, Independent Researcher main coauthors Famiani and Leegood

·

Basic reporting

-Comment 1: Abstract : “was highest in all members” This seems incorrect - Please modify.
-Comment 2: Abstract “saltalkali” Typo.
-Comment 3: Figure 9 « upriht stem » Typo
-Comment 4: Despite the author efforts to review the manuscript language, mistakes and typos are still present throughout the manuscript. Once again, please carefully reread/review the entire manuscript accordingly, before further re-submission.

Experimental design

-Comment 5: Supp Fig 9: Please modify VmSWEET gene names using the updated nomenclature in the manuscript. Name VmSweet in all figures accordingly.
-Comment 6: Supp Fig 10: Please modify VmSWEET gene names using the updated nomenclature in the manuscript. Name VmSweet in all figures accordingly.

Validity of the findings

-Comment 7: Abstract “In the conserved domains, we observed seven phosphorylation sites and four amino acid residues that were crucial for the binding function”
Please review this sentence. The authors would require further experimental evidence to show that phosphorylation sites and amino acid residues are crucial for the binding functions (ie. binding experiment and mutant experiments would be required to state this). Please modify accordingly. Please stress on the speculative aspect of your statement.

-Comment 8: line429 VmSWEET5 in Clade IV may control the flux of fructose across the plasma membrane. The VmSWEET in clade IV is named VmSWEET16 (not 5).

-Comment 9: line 545-551: “For instance, VmSWEET14 and VmSWEET16 were highly expressed during the young fruit stage andexpansion stage, respectively, whereas VmSWEET4 was highly expressed during the color change and the maturity stage. We speculate that VmSWEET4 positively regulates fruit development and ripening, while VmSWEET14 and VmSWEET16 may play roles that are similar to with SlSWEET7a and SlSWEET14. Suppressing the two genes could be a potential strategy forenhancing the sugar content of cranberry fruits.”
This part seems too speculative. I would remove this part.

Reviewer 3 ·

Basic reporting

The basic paper is very much improved.

Experimental design

I think more detail is still required for some of the data collection and analyses. I have outlined some in the attached document.

Validity of the findings

The findings are mostly computational, which is fine. I do question the accuracy and calculations associated the gene expression work. The issues and steps associated with entire process is such that a certain amount of variation is inherent in the process. I am very skeptical of significance of differences at levels much less than 1 or 2 fold.
From a publication:
We consider a RQ significant when there is a minimum of two-fold change: RQ of more than 2 or less then 0,5. This is within the variations of the technique. You might get a RQ of 0.8 one day and easily get one of 1.2 the next day. If you are looking for a small fold (1.5-fold), you need perfect samples very very clean, do technical quadruplicates, consider using 2 assays for this gene, and use positive samples with a known fold-change.

Additional comments

I tried to point out most of the remaining issues in the attached. While many issues have already been addressed, another review may be needed.

Annotated reviews are not available for download in order to protect the identity of reviewers who chose to remain anonymous.

---

## Round 0.3 · Minor Revisions

Dear authors,

I have reviewed your manuscript post its recent re-submission.

There are a number of textual issues which require addressing before your MS can be considered further.

The Discussion section requires the most work, so please take the time to rework this section of the MS.

Please also address all issues that I have identified on the annotated PDF of your manuscript.
Kind regards,
Andy

---

## Round 0.4 · accepted · Accept

Dear authors,
Thank you kindly for making the changes requested as part of this round of review.
I have been through the latest version of your manuscript and I believe it is now at a standard that can be accepted for publication.
Well done on having your study accepted for publication in PeerJ.
Kind regards,
Andrew Eamens